# Indole-Based Tubulin Inhibitors: Binding Modes and SARs Investigations

**DOI:** 10.3390/molecules27051587

**Published:** 2022-02-28

**Authors:** Sheng Tang, Zhihui Zhou, Zhiyan Jiang, Wufu Zhu, Dan Qiao

**Affiliations:** School of Pharmacy, Jiangxi Science & Technology Normal University, Nanchang 330013, China; shengtang_1219@163.com (S.T.); zhouzhihui8412@163.com (Z.Z.); jiangzhiyan9911@163.com (Z.J.); qiaodan2021520@163.com (D.Q.)

**Keywords:** tubulin inhibitors, indole, cancer, binding modes, SARs investigations

## Abstract

Tubulin inhibitors can interfere with normal cell mitosis and inhibit cell proliferation through interfering with the normal structure and function of microtubules, forming spindle filaments. Indole, as a privileged pharmacological skeleton, has been widely used in anti-cancer inhibitors. A variety of alkaloids containing an indole core obtained from natural sources have been proven to inhibit tubulin polymerization, and an ever-increasing number of synthetic indole-based tubulin inhibitors have been reported. Among these, several kinds of indole-based derivatives, such as TMP analogues, aroylindoles, arylthioindoles, fused indole, carbazoles, azacarbolines, alkaloid nortopsentin analogues and bis-indole derivatives, have shown good inhibition activities towards tubulin polymerization. The binding modes and SARs investigations of synthetic indole derivatives, along with a brief mechanism on their anti-tubulin activity, are presented in this review.

## 1. Introduction

Cancer, a multifaceted, multi-mechanistic and life-threatening disease, is one of the appalling scourges worldwide. Cancer cells are characterized by uncontrollable proliferation, unstoppable differentiation and unpredictable migration [1,2,3]. Modern medicine has made great progress in the diagnosis and treatment of cancer. Although many small molecule drugs have been put into clinical treatment, the complete cure of cancer remains a challenge. Over the last decades, medicinal chemists have played a vital role in the exploration, design and synthesis of drugs. In order to find optimal compounds that could serve as initial hit molecules, a large group of compounds based on the structure of the leading drug for biological evaluation in the screening process, along with the use of molecular docking for secondary proof, were designed and synthesized [4,5,6].

Indole is a monocyclic heterocycle compound that consists of a benzene ring and a *N*-containing pyrrole ring. It is a colorless crystal with a pungent odor that reddens readily upon exposure to sunlight and air [7]. The unshared electron pair of the nitrogen atom in the pyrrole molecule participates in the conjugation of the ring system, and so it weakens the binding ability with H and does not show strong alkalinity. Indole analogues were widely distributed in natural products and its derivatives, such as vinblastine isolated from *Catharanthus roseus*, which has been employed in the treatment of various cancers [8]. Since the last century, several indole derivatives have been provided as a versatile pharmacological structure in drug modification. These derivatives have been widely used in target-based anti-cancer inhibitors, and some of which have been approved by the FDA (Table 1).

Tubulin is a globular protein that assembles to form microtubules by a process termed as tubulin polymerization. Microtubules have been an important therapeutic target in tumor cells [9,10], and play an indispensable role in maintaining the cell shape, regulation of motility and cytoplasmic transport [11,12]. Therefore, inhibiting the microtubule function could prevent chromosome segregation at the time of cell division, thus obstructing cancer cell proliferation [13].

Over the years, indole or a 1*H*-benzo[*b*]pyrrole scaffold has been proven as a tubulin polymerization inhibitor. Vincristine, a tubulin drug, is an anti-tumor alkaloid of *Catharanthus roseus*, which inhibits microtubule formation in the mitotic spindle and leads to splinter cell stagnation in the metaphase. Its binding Ki with the microtubule is 85 nm [14]. Molecular docking simulation shows that the binding site of vincristine (Col, cyan) was in the cavity between the β1-subring and α2-subring of tubulin. The indole-skeleton of vincristine has a hydrogen bond interaction with ASN329 and a π-σ conjugation with PRO325 [1]. As a core pharmacophore, the indole scaffold also served a crucially important role in interacting with the colchicine binding site (Col, violet), which has a small cavity volume and is easier to modify. The docking study showed that CYS241 forms a dihydrogen bond with two methoxy groups from the benzene ring of colchicine. In addition, MET259 and LEU255 have π-σ and π-sulfur interactions with colchicine, respectively (Figure 1, PDB:1Z2B) [15]. Accordingly, we focused on the colchicine binding site of tubulin and current application of indole-derived compounds in the design of new anticancer agents. This review summarizes several important substituted indole classes, such as trimethoxyphenyl (TMP) analogues, aroylindoles, arylthioindoles, fused indole, carbazoles, azacarbolines, alkaloid nortopsentin analogues and bis-indole derivatives, through binding modes and SARs investigations studies (all 3D diagrams of the docking analysis below were performed using AutoDock 4.2 software (The Scripps Research Institute, USA), followed by the docking results, which were processed and generated by PyMOL 2.x software (https://pymol.org, accessed on 11 August 2020). Among them, the yellow dashed lines represent the H-bond, residues that have a hydrogen bond interaction with ligands are represented in pink and residues that have a hydrophobic interaction with ligands are represented in violet), with a view of discovering diverse new anticancer agents.

## 2. Indole-Based Tubulin Inhibitors

### 2.1. TMP Analogues

Combretastatin A-4 (CA4), a natural product isolated from *Combretum caffrum*, is a potent tubulin inhibitor that targets colchicine-binding sites. CA-4 has antiangiogenic and antitumor activities, but it presents clinical defects, such as poor pharmacokinetic properties and water solubility [16,17]. Similar to colchicine, CA-4 also has a TMP moiety, which is a key pharmacophoric point for tubulin binding [18,19]. The CA-4 analogue has a TMP moiety and an aromatic ring, which are linked by a wide variety of moieties, such as heterocyclic atoms [20,21,22], acrylamide [16], carbonyl groups [23], etc. The TMP moiety of CA-4 (A ring) is an important skeleton for activity, and the aromatic ring (B ring) can be replaced by different groups. As a “privileged group”, the indole skeleton has been widely used in the modification of CA-4 (Figure 2).

A new class of sulfur atom-spaced TMP derivatives **1a-1l** [24] were designed and synthesized bearing a heterocyclic ring at position 5-, 6- or 7- of the indole nucleus. The structure–activity relationship analysis revealed that 6- and 7-heterocyclyl-1H-indole derivatives, especially compound **1k**, showed a potent inhibition of tubulin polymerization (IC_50_ = 0.58 ± 0.06 µM), binding of colchicine to tubulin and growth of MCF-7 cancer cells, with an IC_50_ value of 4.5 ± 1 nM. The molecular docking of tubulin structures (PDB code: 1SA0) revealed that the five-membered heterocyclic ring at the benzene ring did not change the binding mode of the tubulin inhibitors: (i) the TMP group established an H-bond with CYSβ241; (ii) the TMP group and indole had hydrophobic interactions with LEUβ248, LEUβ255, ASNβ258 and METβ259; (iii) the five-membered heterocyclic ring was located in the cavity of the colchicine binding site and had hydrophobic interactions with Metβ259, Lysβ353, Alaα180 and VAL181. Wen et al. [25] developed a facile and effective microwave-assisted synthesis of a series of selenium atom-spaced TMP derivatives (**2a-2q** and **3a-3l**), and their selenoxides derivatives were designed as a new class of tubulin inhibitors. Most of the derivatives exhibited significant anti-proliferative activity, especially derivative **3a**, with IC_50_ values of 12.3 ± 1.6, 13.5 ± 1.5 and 25.1 ± 2.0 nM against three cancer cells (SGC7901, KB and HT1080), respectively. The in vitro tubulin polymerization combined with the immunofluorescence assay proved that the mechanism of action of compound **3a** is to disrupt tubulin microtubule dynamics by inhibiting tubulin polymerization. In addition, in order to investigate the possible binding of target compounds to the colchicine site of tubulin, a docking study was carried out on the most effective derivative: **3a**. The docking results (PDB code: 1SA0) showed that compound **3a** was tightly bound to the tubulin binding site. The oxygen atom of the 4′-methoxy group of **3a** and the sulfhydryl group of CYS241 establish a key strong hydrogen bond. The results of this docking study were highly consistent with the strong antiproliferative activity of **3a** and its ability to inhibit tubulin polymerization. Romagnoli et al. [26] synthesized a new class of indole-based TMP inhibitors with a nitrogen spacer (**4a-4j** and **5a-5s**) of tubulin polymerization and evaluated their in vitro anti-proliferative activity, inhibition of tubulin polymerization and cell investigation. By exploring the structure–activity relationship, it was found that the substitution of the C-6 methoxy group on the indole nucleus plays an important role in inhibiting cell growth. Therefore, the derivatives **5f** and **5m** were identified as the most promising compounds, and exhibited potent inhibitory activity against the four cancer cells, with IC_50_ values ranging from 0.11 to 1.4 μM, and compound **5m** could effectively inhibit tubulin polymerization (IC_50_ = 0.37 ± 0.07 μM). In addition, the molecular docking study (PDB code: 4O2B) showed that the series of derivatives could bind to the colchicine binding site on tubulin. Among them, the 4′-methoxy group in the 3′,4′,5′-trimethoxyphenyl group of the compound **4f** formed a hydrogen bond interaction with amino acid CYS241, while there was an interaction between the indole NH and THR179 and between the carboxylic group and ALA250/ASN268. Moving the methoxy group from position 5 to 6 (**5m**) could correctly occupy the binding area, which was equivalent to the binding position of **4f**. The 6-methoxy group and the carboxyl group (C=O) formed a hydrogen bond with amino acid residues MET259 and ALA250, respectively. Further mechanism studies showed that the compound **5m** could induce apoptosis of HeLa cells and cause cell cycle arrest in the G2/M phase (Figure 3). 

Chen et al. [27] designed and synthesized a series of indole-based TMP analogues with an amide spacer (Figure 4A) and evaluated their anti-proliferative and tubulin polymerization inhibitory activity. Most of the derivatives demonstrated significant anti-proliferative activity against six human cancer cells. Among them, the compound **6v** exhibited the most potent inhibitory activity against the T47D cells, with an IC_50_ value of 0.04 ± 0.06 µM. A flow cytometric analysis showed that the compound **6v** significantly inhibited the growth of breast cancer cells through arresting cell cycle in the G2/M phase in a dose-dependent manner. Importantly, the compound **6v** also exhibited the most effective anti-tubulin activity, with an IC_50_ value of 9.5 µM (Figure 4A). To investigate the possible binding mode for this series of compounds, a molecular docking simulation was performed on the binding site of the most active compound, **6v**, and tubulin colchicine (PDB code: 5LYJ). As shown in Figure 4B, the TMP group in proximity to CYSβ241 and the chlorobenzene ring of the indole backbone set up hydrophobic interactions with the LYSβ254, VALα181, ASNβ258 and METβ259 side chains. In addition, three hydrogen bonds were observed: (i) the 1-NH of the indole ring donates a hydrogen bond to THRα179; (ii) the carbonyl group of the 4-chlorobenzoyl moiety accept a H-bond from ASNβ258L; (iii) one of the oxygens of the TMP group forms a hydrogen bond with CYSβ241.

Liou et al. [28] firstly introduced an anti-tubulin agent, BPR0L075, by adding a ketone structure between the indole nucleus and TMP skeleton on the bases of bioisosterism, where it showed excellent anti-proliferative activity against several human cancer cells, with IC50 values of 1–23 nM. The emergence of BPR0L075 caught the attention of researchers. Mullagiri et al. [29] prepared indole–benzimidazole conjugates with a carbonyl substitution bridge and conducted in vitro cytological investigations against four human cancer cells. The preliminary activity screening showed that most of the derivatives showed significant inhibitory activity, especially compounds **7g** and **8f**, which exhibited potent inhibitory activity against human prostate cancer cells DU-145, with IC50 values of 0.68 and 0.54 µM, respectively (Figure 5A). Further study revealed that these compounds arrested the cell cycle at the G2/M phase and induced apoptosis in DU-145 cells. Molecular modeling studies (PDB code: 5LYJ) showed that the TMP group of the compound **7g** accepts two hydrogen bonds from SER140 and GLN11. Additionally, the -NH group of benzimidazole ring donates a hydrogen bond to THR179. It is worth noting that the carbonyl group at the bridge chain has a hydrogen bond interaction with CYS254, which was rarely reported in this class of tubulin inhibitors. The interactions with other active site residues GLN247, LEU248, LEU255, ASN258, MET259 and LYS352 were also observed. However, in compound **8f**, the para and meta-positions of the TMP group established hydrogen bonds with CYS241.

Wang et al. [30] innovatively introduced a benzimidazole skeleton between the TMP group and indole group as a potential tubulin polymerization inhibitor. In vitro cell viability experiments showed that compound **9** had the most potent tubulin polymerization inhibitory activity (IC_50_ = 1.5 ± 0.56 μM), and exhibited anti-proliferative activity against A549, HepG2 and MCF-7 cells, with IC_50_ values of 2.4 ± 0.42, 3.8 ± 0.5 and 5.1 ± 0.42 μM, respectively (Figure 5). The mechanism action study confirmed that compound **9** could effectively induce the apoptosis of the A549 cell associated with G2/M phase cell cycle arrest. The docking results (PDB code: 1SA0) revealed that two oxygens from the C-5 methoxy group on the indole nucleus and TMP groups had hydrogen bond interactions with CYS241 and LEU248, respectively. In addition, LYS352 and LEU248, located in the binding pocket of tubulin, had a Pi-cation interaction with compound **9**, which increased the stability of the structure. (Figure 5B).

Yan et al. [31] synthesized a new class of indole-based TMP analogues (**10a-10l**), with a Michael receptor as a spacer. The detailed SARs of these compounds are shown in Figure 5. In vitro anti-proliferative activity demonstrated that compound **10k** showed the strongest inhibitory activity, with IC_50_ values of 3–9 nM against four human cancer cells, as well as a high selectivity to normal cells and tubulin polymerization inhibitory activity (IC_50_ = 2.68 ± 0.15 μM). Further immunofluorescence assays and molecular docking studies showed that compound **10k** was a novel tubulin polymerization inhibitor (Figure 5). The cellular mechanism study clarified that the compound **10k** arrested the A549 cell cycle at the G2/M phase and induced apoptosis. In addition, compound **10k** not only showed good metabolic stability in mouse liver microsomes, but also inhibited tumor growth in the in vivo xenograft model without significant toxicity.

### 2.2. Arythioindoles

Sulfanilamide and its derivatives have a wide range of pharmacological effects, such as antibacterial, anti-inflammatory, antiviral and anti-parasitic [32,33]. In addition, several sulfonamides have been reported to show significant anticancer activity in vitro and in vivo, and some of these compounds are currently undergoing clinical trials [34,35,36]. Most of these compounds contain aromatic or heterocyclic sulfonamide groups, and they involve multiple antitumor mechanisms, such as the disruption of tubulin assembly. Aceves et al. [37] synthesized a series of derivatives (**11a-11b** and **12a-12f**) containing sulfonamide scaffolds (Figure 6) and evaluated them for their in vitro anti-proliferative activity against representative human cancer cells, thus screening out the derivative **12e**, which displayed clear anti-proliferative properties. A cell cycle distribution analysis revealed that compound **12e** could cause the arrest of the cell cycle in the G2/M phase and induce apoptosis. Tubulin staining showed that compound **12e** inhibited the polymerization of tubulin in vitro in a dose-dependent manner, and was also able to strongly inhibit the cancer cell motility.

Zhou et al. [38] introduced a N atom to the five-membered ring to replace the sulfonyl group based on the lead compound **E28** [39], a STAT3 inhibitor, and synthesized a class of N-arylsulfonylsubstituted-1*H* indole derivatives, **13a-13x**, as dual STAT3 and tubulin inhibitors. The anti-proliferative activity showed that most of the derivatives displayed strong inhibitory activity in vitro. Among them, the derivative **13a** was identified as the most potent compound (Figure 6), which not only retained the inhibitory activity toward STAT3 phosphorylation but also empowered the inhibitory ability toward tubulin polymerization. Meanwhile, the IC_50_ values of compound **13a**’s anti-proliferative activity against a variety of cancer cells were less than 10 μM (Figure 6). Similar to colchicine, compound **13a** inhibits tubulin polymerization in a concentration-dependent manner. Further bioactivity assays revealed that compound **13a** arrested the A549 cell cycle at the G2/M phase.

Based on bioelectronics, Li et al. [40] replaced the vinyl carbonyl group of the CA-4 analogue with vinyl sulfoxide as Michael acceptors and afforded a class of novel indole-vinyl sulfone analogues **14a-14p** as tubulin polymerization inhibitors (Figure 7A). The results of the in vitro anti-proliferative activity showed that the most of the derivatives improved the metabolic stability and drug-likeness properties. In particular, compound **14e** exhibited the most potent activity against a panel of human cancer cells, with IC_50_ values of 55–305 nM, and might have had a lower toxicity to normal LO2 cells (IC_50_ = 0.240 ± 0.090 μM) (Table 2). In addition, compound **14e** inhibited microtubule polymerization by binding to the colchicine site of tubulin. Further mechanistic studies demonstrated that compound **14e** caused cell cycle arrest at the G2/M phase, induced cell apoptosis and disrupted microtubule networks in K562 cells in a dose-dependent manner. Importantly, compound **14e** effectively inhibited tumor growth without significant toxicity in the in vivo H22 liver cancer xenograft mouse model. The molecular docking results (PDB code: 1SA0) showed that the TMP ring of **14e** was located in a hydrophobic pocket of the colchicine binding cavity, that one oxygen atom of the TMP group formed a hydrogen bond with the critical residue CYS241 andthat the indole ring reached into another hydrophobic pocket in the opposite direction, which is consistent with the tropone of colchicine. (Figure 7B).

### 2.3. Aroyindoles

The imidazolo [1, 2-α] pyridine (IMPs) skeleton is a special rigid heterocyclic ring skeleton for medicinal use that has special application value in the generation of small molecule ligands with obvious anti-tumor, anti-viral and anti-inflammatory effects [41]. As mentioned above, the antitumor activity of CA-4 is often reduced by changes in the *cis*-olefin configuration at the linking bridge. To develop potent anti-tubulin agents targeting the colchicine-binding site, Zhai et al. [42] introduced IMPs at the linking bridge and designed a series of novel 3-aroyindoles derivatives **15a-15v** by bioisosterism. Zhai et al. effectively synthesized these novel derivatives via an acylation reaction, one-pot coupling reaction and cyclization transformation, and evaluated the inhibitory activity against five human cancer cells and normal human cells (LO2) (Figure 8A). The compound **15k** was screened by a structural–activity relationship, which not only exhibited significant anti-proliferative activity against all tested human cancer cells, with IC_50_ values of 20–1220 nM, but also had little cytotoxic effect against LO2 cells. The cell cycle study showed that compound **15k** could lead to G2/M cell cycle arrest in vitro. The docking study (PDB code: 4O2B) indicated that compound **15k** formed two key H-bonds with VALβ181 and ASPβ251 of β-tubulin, as shown in Figure 8B. The IMP skeleton could link up to METβ259, ALAβ316, LYSβ352 and LEUβ255 by hydrophobic interaction, suggesting that the IMP group at position 3 of the indole scaffold showed excellent tubulin-binding potency. In addition, the indole moiety reached into the hydrophobic pocket formed by LEUβ248, CYSβ241, LEUβ255 and ALAβ250.

Inspired by the pharmacophore CA-4, Liu et al. [43], developed a series of novel 3-aroyindoles derivatives **16a-16n**, similar to the work of Zhai et al. The preliminary anti-proliferative activity showed that most of the derivatives displayed promising inhibitory activity against five selected human cancer cells, with IC_50_ values in a double-digit nanomolar range. Among them, the most promising compound, **16e**, exhibited excellent inhibitory activity against HT29 and H460 cells, with IC_50_ values of 0.01 ± 0.002 and 0.04 ± 0.006 μM, respectively (Figure 8A). Molecular docking (PDB code: 4O2B) showed that multiple amino acid residues stabilize compound **16e** at the colchicine binding site, and that the substituents on benzimidazole have hydrogen bonding and hydrophobic interactions with several key amino acid residues in the cavity. Concretely, the diflourophenyl moiety extended into the hydrophobic pocket formed by ALA250, THR179, ASN101, GLY143 and SER178. In addition, the nitrogen on the cyanide group had a H-bond with ASN249. The indole skeleton and cyclohexyl group have Pi-sulfur and alkyl interactions with several amino acids, which are depicted in Figure 8C.

Kazan et al. [44] synthesized a class of indole-2-carbohydrazides (**17a-17e** and **18a-18h**) as potent tubulin polymerization inhibitors, and evaluated their anti-proliferative activity. A preliminary screening was conducted for compounds **17a** and **17b** against 60 human tumor cells at a minimum of five concentrations at 10-fold dilutions. The results showed significant anti-proliferative activity against leukemia and non-small cell lung cancer. By structural modification and a structure–activity relationship study, compounds **18f** and **18g** exhibited potent anti-proliferative activity against MCF-7 cells, with IC_50_ values of 0.42 ± 0.06 and 0.17 ± 0.02 μM, respectively (Figure 9A, and displayed significant inhibitory activity on both tubulin assembly (with IC_50_ values of 1.7 ± 0.06 and 1.4 ± 0.02 μM, respectively) and colchicine binding. Simultaneously, the molecular docking study showed that the carboxamide oxygen of compound **18g** had electrostatic interactions with LYSβ254, while the indole ring occupied the space at the interface of the two monomers. The substituted phenyl moiety was embedded into the hydrophobic pocket formed by several amino acid residues, such as PHEβ169, TYRβ202, VALβ238, CYSβ241, LEUβ242, LEUβ248, ALAβ250, LEUβ252, LEUβ255, ALAβ316 and ILEβ378 of β-tubulin. 

In addition, 2, 3-diarylindole derivatives have been reported as inhibitors of tubulin polymerization [45]. Thanaussavadate et al. [46,47] designed and synthesized a FITC fluorophore via a hydrophilic linker to 2,3-diarylindole derivatives **19a-19d** (Figure 9A) and studied their biological activity against non-small cell lung cancer cell lines. Through the structure–activity relationship study, the optimal position of the linker was determined to obtain the compound **20** with the most excellent inhibitory activity against A549 cells (IC_50_ = 5.17 ± 1.61 μM) and could inhibit the polymerization of tubulin in vitro. The molecular docking study (PDB code: 1SA0) showed that compound **20** could bind to the colchicine binding site of tubulin (Figure 9B).

### 2.4. Fused Indole

In recent years, several research groups have introduced a fused indole ring system in the design of novel tubulin inhibitors. The *cis*-double bond of CA-4 is easy to isomerize, and a ring structure is introduced in the position of the double bond to maintain its biological activity. Inspired by the colchicine structure, Yan et al. [48] replaced the *cis*-double bond with seven-membered rings and developed a series of fused indole derivatives as tubulin polymerization inhibitors. In vitro anti-proliferative activity showed that all of the derivatives not only displayed effective anti-proliferative activity, but also exhibited excellent inhibition activity of tubulin polymerization. Among them, compound **21** exhibited significant antiproliferative activity against several human cancer cells, with IC_50_ values ranging from 22 to 56 nM, and also inhibiting tubulin polymerization (IC_50_ = 0.15 ± 0.07 µM) (Figure 10). The immunofluorescence assay revealed that the optimal compound, **21**, disrupted the intracellular microtubule network and interfered with cell mitosis. The cellular mechanism study showed that compound **21** arrested the cell cycle at the G2/M phase and induced apoptosis in a time- and dose-dependent manner.

Inspired by the natural product evodiamine [49], a series of indolopyrazinoquinazolinone derivatives were designed by scaffold hopping (Figure 10) [50]. The structure–activity relationship study showed that compound **22** showed low nanomolar inhibitory activity against the HCT116 cells, with an IC_50_ value of 2 nM (Figure 10). A further mechanism study indicated that compound **22** could inhibit topoisomerase 1 (Top1) and tubulin, and induced apoptosis with G2 cell cycle arrest. In the low toxicity HCT116 xenograft model, the quaternary ammonium salt of compound **22** exhibited excellent in vivo inhibitory activity (TGI = 66.6%). The docking results revealed the interactions of compound **22** with tubulin and topoisomerase 1. Compound **22** showed a rare “planar” spatial structure after energy minimization and established base stacking interactions with the base residue TGP11 in the active pocket of topoisomerase 1 (PDB code: 1T8I). Meanwhile, compound **22** occupied well in the colchicine-binding cracks of tubulin. The oxygen of the B-ring accepted a H-bond from ALA250, and the A-ring established a salt bridge with the adjacent LYS254 (PDB code: 1SA0).

### 2.5. Carbazoles

Several studies have shown that carbazole derivatives have significant biological activities, such as anticancer, psychotropic, anti-inflammatory, anti-histaminic and anti-biotic activities [51,52,53,54,55,56,57,58]. Mahanine [58] and ellipticine [59] are two carbazole-based nitrogen heterocycle products extracted from nature with excellent anti-cancer activity. Based on the structural modification of lead compound **IMB105** [60], Liu et al. [61] designed and synthesized novel *N*-acyl- and *N*-alkyl-substituted carbazole sulfonamide derivatives (**23a-23x**) and evaluated their anti-proliferation efficacy and water solubility (Figure 11). The results showed that most of the derivatives displayed good aqueous solubility and presented strong in vitro anti-proliferation and in vivo inhibitory activity. The structure–activity relationship showed that the compound **23v**, with N-ethyl phosphate sodium moiety, exhibited the strongest inhibitory effect against HepG-2, Bel-7402 and MCF-7 cells, with IC_50_ values of 1.12 ± 0.22, 1.97 ± 0.76 and 1.08 ± 0.23 µM, respectively (Figure 11). In addition, they also developed a novel synthetic method for 7-hydroxy substituted carbazole sulfonamide compound **24a** [62] and sodium phosphate prodrug **24b** and investigated their in vivo inhibitory activity. The results showed that the compound **24b** significantly inhibited the growth of tumor in the mouse HepG2 xenograft model without observed toxicity, and showed good pharmacokinetic properties.

### 2.6. Azacarbolines

Carbolines are another vital skeleton of natural and synthetic indole alkaloids. Carbolines, such as α- and *β*-carboline derivatives, have been proven to have potent antitumor activity, among which, *β*-carboline has outstanding therapeutic and physiological significance. Azacarbolines have significant pharmaceutical benefits in the field of cancer due to their biological isosteric bodies, which are similar to *β*-carbolines [63,64,65,66,67,68,69]. Samundeeswari et al. [70] synthesized a series of coumarin tetrahydro-*β*-carboline hybrids **24** by the Pictet–Spengler reaction, and then oxidized them by DDQ to obtain dihydro hybrids **25** and *β*-carboline hybrids **26**. The in vitro anti-proliferative activity of all derivatives was evaluated against 60 cells, and the compound **26c** was screened for further molecular docking study. The compound **26c** formed four H-bonds with three amino acid residues at the colchicine binding domain: (i) ASP251 formed a bidentate hydrogen bond with two oxygens of the coumarin ring; (ii) LEU252 donated a H-bond to the oxygen on a carbonyl group; (iii) ALA 250 had a hydrogen bond interaction with the oxygen of the coumarin ring (PDB code: 1SA0). In addition, compound **26c** interacted with the hydrogen of ASN29 through the nitrogen of β-carboline, and only interacted with one amino acid, but did not interact with coumarin (PDB code: 1Q08). The results showed that derivative **14c** had binding properties to both the kinesin spindle protein (KSP) and tubulin (Figure 12).

### 2.7. Alkaloid Nortopsentin Analogues

Nortopsentins, a kind of bis-indolyl alkaloid, are isolated from the Caribbean deep sea *Spongosorites ruetzleri*. Nortopsentin analogues have a five-membered heterocyclic ring (imidazole) as a spacer between the two indole units. The study shows that nortopsentin analogues display in vitro antiproliferative activity [71,72]. In order to develop new anti-cancer drugs, Yang et al. [73] introduced a five-membered heterocyclic pyrazolidine at the bridge link and obtained a series of nortopsentin derivatives, **2****7a-27v**, as tubulin inhibitors (Figure 13A). The results of in vitro anti-proliferative activity showed that compound **2****7q** not only displayed the strongest in vitro inhibitory activity against A549, MCF-7 and HepG2 cells, with IC_50_ values of 0.15 ± 0.03, 0.17 ± 0.05 and 0.25 ± 0.05 µM, respectively, but also exhibited significant tubulin polymerization inhibitory activity, with an IC_50_ value of 1.98 ± 0.25 µM. The mechanism of action studies proved that compound **2****7q** could potently induce apoptosis in A549 cells and cause the arrest of the cell cycle in the G2/M phase. The molecular docking study (PDB code: 1SA0) showed that compound **2****7q** was well occupied in the active pocket via hydrophobic interaction. The oxygen of the acetyl group substituent on pyrazolidine contributed a hydrogen bond acceptor and had a H-bond interaction with the N-H of residue ALA250. (Figure 13B). In addition, residues ALA 250, MET259, CYS241, LEU255, LEU248 and LYS352 were predicted to make contact with colchicine or compound **2****7q**.

In 2016 and 2018, Zhang et al. [74,75] successively reported a series of nortopsentin analogues (**28 series** and **29a-29y**) with a substituted pyrazole ring as a spacer (Figure 13A). The in vitro cell inhibitory activity assay and structure–activity relationship indicated that most of the derivatives showed potent anti-proliferative activity. Among them, the derivatives **28** and **29x** were identified as the most desired compound, respectively. Not only did they exhibit the most effective activity against four human cancer cells, where the latter showed no toxicity to normal human cells, but they also exhibited the most potent inhibitory effect against tubulin assembly. The cellular mechanism study clarified that compounds **28s** and **29x** caused mitotic arrest in the G2/M phase and disrupted the microtubule system. The molecular docking study (PDB code: 1SA0) showed that compound **28s** was well occupied into the colchicine binding domain of tubulin via hydrophobic interactions, and was stabilized by two amino acid residues. The oxygen of the acetyl group substituent on the pyrazole ring contributed a H-bond with the hydrogen of the hydroxy group of SER178. Meanwhile, the oxygen of the methoxy group on the benzene ring had a hydrogen bonding interaction with ASN178 (Figure 13B). Regarding compound **29x** (PDB code: 1SA0), the m-methoxy phenyl maintained the H-bond interaction with CYSβ241 and was surrounded by hydrophobic residues, such as LEUβ248, LEUβ255, ALAβ316, ALAβ317, VALβ318, ALAβ354 and ILEβ378. The carbonyl group (C=O) formed another hydrogen bond interaction with the amino acid residue ASNβ258. Importantly, compound **29x** could significantly inhibit tumor growth in HeLa-xenograft nude mice, with a relative tumor rate of up to 61.52%, without noticeable weight loss and tissue damage ASN178 (Figure 13B).

### 2.8. Bis-Indole Derivatives

Indirubin (**30**) is a new type of bis-indole alkaloid, found in Chinese herbal medicine Indigo, that has a therapeutic effect on chronic myeloid leukemia (CML) [76,77]. Lakshmi et al. [78] predicted that indirubin might inhibit the polymerization of tubulin and, therefore, deciphered the underlying molecular mechanism of indirubin (**30**) antitumor properties. The anti-cancer cell proliferation activity indicated that indirubin (**30**) against HeLa cells had an IC_50_ of 40 μM and could inhibit the proliferation of L929 cells by 53% at a concentration of 20 nM. Immunofluorescence microscopy showed that indirubin (**30**) could misalign chromosomes on the metaphase plates of HeLa cells at 20 and 40 μM concentrations, suggesting that indirubin (**30**) disrupted microtubule dynamics. When the concentration was increased to 80 μM, indirubin significantly depolymerized the spindle and misaligned the chromosome of HeLa cells. Like other tubulin destabilizers, indirubin (**30**) disrupts microtubule dynamics at low concentrations, whereas it induces microtubule depolymerization at higher concentrations. The molecular docking of indirubin (**30**, PDB code: 1SA0) revealed that indirubin was located in a new binding interface of the α_1_-β_1_ tubulin heterodimer, which was close to the colchicine binding site (Figure 14A). Oxygen atoms on two carbonyls groups had hydrogen bonding interactions with residues LYS326, ARG214 and GLU220. Meanwhile, bis-indole rings extended into a hydrophobic pocket formed by residues GLN 176, TYR 210, ARG 221, PRO 222, MET 325, ASP 329, GLU 330 and LEU 333 (Figure 14B).

Based on previous studies [79], a bis(indolyl)-hydrazide-hydrazone derivative NMK-BH2 (**31**) [80] was identified as a new anti-microtubule drug, and further research was conducted. The results of in vitro anti-proliferative activity showed that compound **31** exhibited a potent efficacy and selectivity against HeLa cells, with an IC_50_ value of 1.5 ± 0.25 µM, whereas it is negligibly toxic towards normal human cells (Table 3). The exploration and characterization of the anti-cancer mechanism mediated by this derivative revealed that it perturbed the cytoskeletal and spindle microtubules of HeLa cells, leading to mitotic blockage and cell death by apoptosis and autophagy. The molecular docking results (PDB code: 1Z2B) showed that the distance between the amide hydrogen atom of the Asn101 residue (of α-tubulin) to the oxygen and nitrogen atom of the pharmacophore linker and the oxygen atom of THR179 (of α-tubulin) to the amide hydrogen atom of the pharmacophore linker enabled the establishment of a conventional hydrogen bond (Figure 15). Therefore, the compound **31** was identified and established as a novel microtubule-targeting anti-cancer drug that could be used as a prospective guide for the development of emerging chemotherapeutic drugs.

### 2.9. Others

The third isomer of the combretastatin A series (*iso*CA) was modified from phenstatin [81], an extremely potent inhibitor of tubulin. Notably, the cytotoxic activity of *iso*Ca-4 was 10 folds that of phenstatin and 4 folds that of colchicine [82]. Li et al. [83] designed and synthesized a series of (*iso*CA-4) analogues by replacing the TMP group with quinoline moiety (Figure 16A). Most of the synthesized compounds showed moderate antitumor activity, among which, compound **32b** exhibited excellent antiproliferative activities against five human cancer cell lines and could effectively inhibit tubulin polymerization, with IC_50_ values of 2.09 μM, which were slightly stronger than that of CA-4 (IC_50_ = 2.12 μM). The cell cycle analysis showed that compound **32b** could block the A549 cell cycle at the G2/M phase in a dose-dependent manner. The molecular docking results (PDB code: 5LYJ) revealed that compound **32b** adopted the colchicine’s binding position. The indole rings extended into the hydrophobic pocket formed by residues THR179, VAL315, ASN350 and VAL351. The hydroxymethyl group of **32b** established two H-bonds with VAL315 and ASN350. The *N*-1 of quinoline moiety formed a key H-bond with CYS241, which was proven to interact with the methoxy group of CA-4. The binding modes of **32b** demonstrated that the quinoline skeleton is an effective surrogate of TMP groups (Figure 16B).

It was reported that a TMP analogue, JAI-51 [84], a kind of microtubule destabilizer, can effectively bind tubulin, and has the same binding site with colchicine. In addition, another microtubule inhibitor, crolibulin, bearing a 3-bromo-4,5dimethoxy-phenyl moiety, was used in clinical trials [85]. Inspired by JAI-51 and crolibulin, Mirzaei et al. [86] designed and synthesized a new class of indole-based chalcone derivatives, **33a-33k**, containing 3-bromo-3,5-dialkoxylated phenyl moiety as tubulin polymerization inhibitors. The in vitro anti-proliferative activity assays against cancer cells revealed that the compound **33b** exhibited the most excellent inhibitory activity against A549 cells, with an IC_50_ value of 4.3 ± 0.2 μM. Further biological evaluations showed that compound **33b** could significantly inhibit tubulin polymerization (IC_50_ = 17.8 ± 0.2 μM) in a dose-dependent manner and decrease the mitochondrial thiol content, thereby inducing the apoptosis of cancer cells. Meanwhile, the binding mode of compound **33b** to tubulin (PDB code: 1SA0) is shown in Figure 17. Compound **33b** occupied the binding pocket of colchicine and was stabilized by four hydrogen bonds. Among them, oxygen at the carbonyl group of the Michael’s receptor accepted three H-bonds from the residues ASP251, LEU252 and LEU255, and had hydrophobic interactions with ALA250 and LYS254 residues. The bromine atom on the phenyl ring formed two key halogen bonds with ALA317 and LYS352 residues. Furthermore, the oxygen on one of the methoxy groups accepted a H-bond from the N-H of ALA316.

In 2007, Kamal et al. [87] reported a series of 2-anilino substituted nicotinyl arylsulfonylhydrazides, **34a-34i**, as a new anticancer agents based on the lead compound **E7010**, which was a tubulin polymerization inhibitor. However, these tubulin inhibitors showed serious side effects in clinical practice. Therefore, in 2015, Kamal et al. [88] designed and synthesized a class of 2-anilinopyridine-arylpropenone conjugates, **35a-35p**, as tubulin polymerization inhibitors, and evaluated them for their in vitro anti-proliferative activity against four different human cancer cells (Figure 18A). All derivatives showed moderate to good anti-proliferative activity against most of the tested cancer cells. Among them, compound **35a** showed promising inhibitory effects against A549, HeLa, MCF-7 and HCT116 cancer cells, with IC_50_ values of 0.51 ± 0.11, 0.65 ± 0.08, 0.71 ± 0.23 and 0.99 ± 0.20 μM, respectively. Meanwhile, compound **35a** displayed significant inhibitory effects of tubulin polymerization, with an IC_50_ value of 1.34 ± 0.3 µM (Figure 18A). Flow cytometry results indicated that compound **35a** could cause cell cycle arrest and accumulated cells in the G2/M phase. In addition, Hoechst staining and the activation of caspase-3 indicated that these conjugates induced cell death by apoptosis. The molecular docking results (PDB code: 3UT5) showed the trimethoxy benzene ring of **35a** showed key hydrogen bond interactions with VALα181. In addition to this, the -NH group of **35a** showed a hydrogen bonding interaction with GLNβ247. Other residues, such as LEUβ252, ASNβ258, VALβ315, ALAβ316, ILEβ318 and ILEβ378, were found to be involved in hydrophobic interactions (Figure 18B).

## 3. Conclusions and Perspective

We screened out 22 ligands embedded in the binding sites of colchicine that had hydrogen bonding interactions with tubulin. Almost half of the ligands were docked with tubulin protein 1SA0 (Figure 19A), a classical structure of tubulin in a complex with colchicine, which was first published in *Nature* by Ravelli et al. [18] in 2004. In addition, 17 residues forming hydrogen bonds with the above 22 ligands were arranged (Figure 19B), among which, residues CYS241 showed the highest frequency (nine times), followed by ALA250 (five times). Most of the ligands that form hydrogen bonds with CYS241 and ALA250 were docked with protein 1SA0. The X-ray structure (1SA0) showed that the CYS241 and ALA250 (red) in β-tubulin were located in the hydrophobic pocket extended by the TMP group of colchicine (Figure 20A). We found that 9 out of 11 ligands docked with protein 1SA0 formed hydrogen bonds with CYS241 or ALA250 and occupied the hydrophobic cavity to which the TMP group of colchicine extends. In addition, all of the TMP analogues described in this paper form hydrogen bonds with CYS241, except the compounds **5m** and **7g**. It is worth noting that Li et al. proved that the quinoline skeleton of compound **32b** was an effective surrogate of TMP groups by showing the H-bond between CYS241 and the quinoline skeleton. All of the above results demonstrated that CYS241 was the key residue of the colchicine binding site and that the cavity where CYS241 and ALA250 are located is the key domain of the colchicine binding site. In addition, we explored the generality that the indole skeleton binds to tubulin. Regrettably, most of the indole rings had substituents on the N atom, such as methyl, which means that there were no extra hydrogen bond donors and acceptors on indoles. The indole moiety of compound **9** (PDB code: 1SA0), compound **4** (PDB code: 4LYJ), compound **6v** and **15k** (PDB code 4O2B) and compound **35a** (PDB code: 3UT5) had hydrogen bonding interactions with the residues LYS352 (1SA0), THR179 (4LYJ), THR179 and ASP251 (4O2B) and GLN247 (3UT5), respectively. The steric sites of the corresponding original ligand and residues mentioned above were shown in Figure 20A–E. Using CYS241 (pink) as a reference, the residues that contain hydrogen bonds with indole are located in the other side of the “L-shaped” cavity, away from the hydrophobic pocket that extends by the TMP group (Figure 20A,B). According to Figure 20C–E, we found that the residues (violet) were located in all directions of the cavity, which meant that the orientation of the indole nucleus was also irregular. However, we noticed that the substitutions on indole N atoms were mostly methyl or small flexible groups. As a result of the rigid backbone of indole, we hypothesized that substituting methyl or small flexible groups on indole would allow for a better extension into the cavity of colchicine.

Tubulin polymerization inhibitors have attracted much attention for their high molecular-targeting specificity and strong anti-tumor activity. Studies have demonstrated that there are three different inhibitor binding sites on tubulin: the paclitaxel site, vincristine site and colchicine site. Due to the small volume of the colchicine site cavity and the relatively simple structure of the corresponding inhibitors, the research on colchicine inhibitors has attracted much attention in recent years.

A great deal of preclinical indole-based tubulin inhibitors have constantly appeared in the literature. This review emphasizes tubulin polymerization inhibitors interacting with the colchicine domain and in vitro cell activity, aiming to obtain ideal tubulin inhibitors and to overcome the common shortcomings of existing tubulin drugs. 

This review also shows some active scaffolds and fragments that play a crucial role in interacting with tubulin and enhancing the inhibition of cell viability in vitro. Currently, there are many limitations of tubulin inhibitors, such as drug resistance, neurotoxicity and bioavailability. Therefore, understanding the mechanisms by which small-molecule inhibitors bind to tubulin is critical to understanding their critical roles in cancer therapy and neurodegenerative diseases. Future research will focus on the synthesis of novel indole analogues that should be highly efficient and should avoid the toxicity of current tubulin-targeting drugs. We expect that this review could provide valuable design ideas for follow-up researchers.

## Figures and Tables

**Figure 1 molecules-27-01587-f001:**
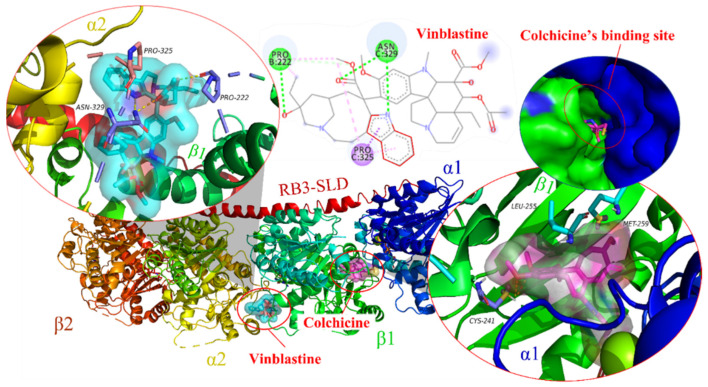
The model of vincristine and colchicine bound to tubulin (PDB code: 1Z2B).

**Figure 2 molecules-27-01587-f002:**
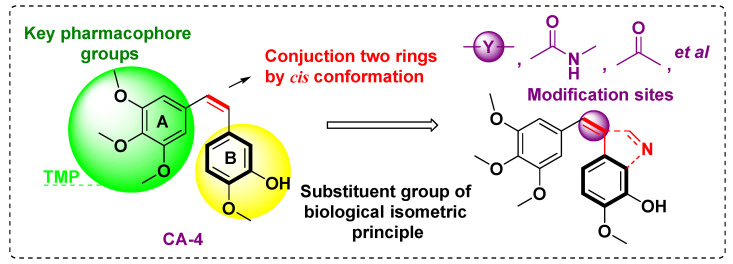
Combretastatin A-4 (CA-4)-based TMP analogues.

**Figure 3 molecules-27-01587-f003:**
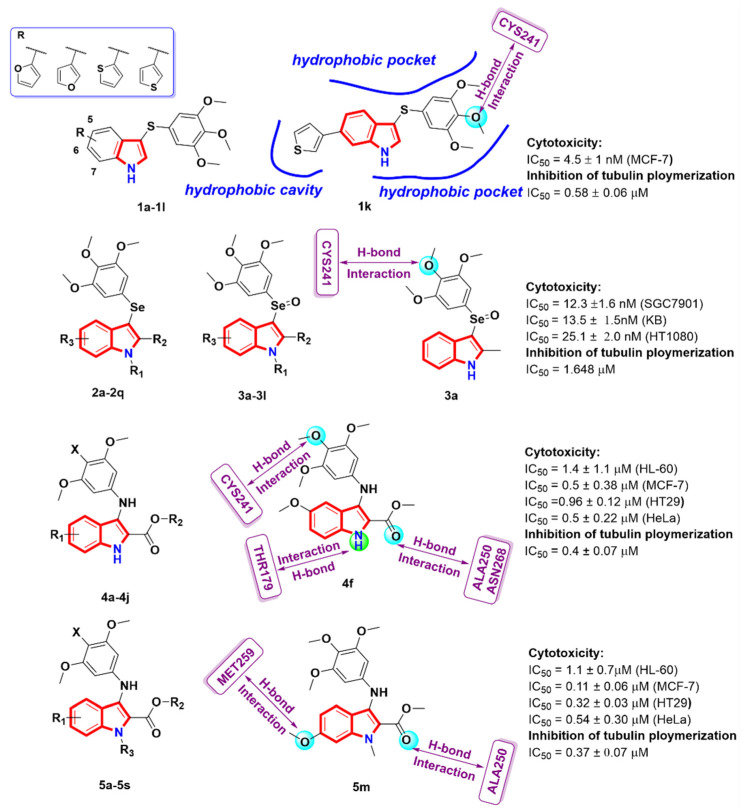
Indole-based TMP analogues.

**Figure 4 molecules-27-01587-f004:**
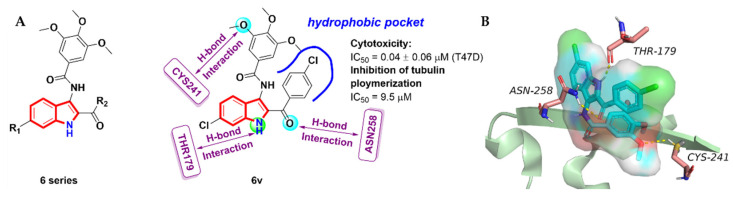
(**A**) Indole-based TMP analogues; (**B**) binding mode for compound **6v** with tubulin (PDB code: 5LYJ).

**Figure 5 molecules-27-01587-f005:**
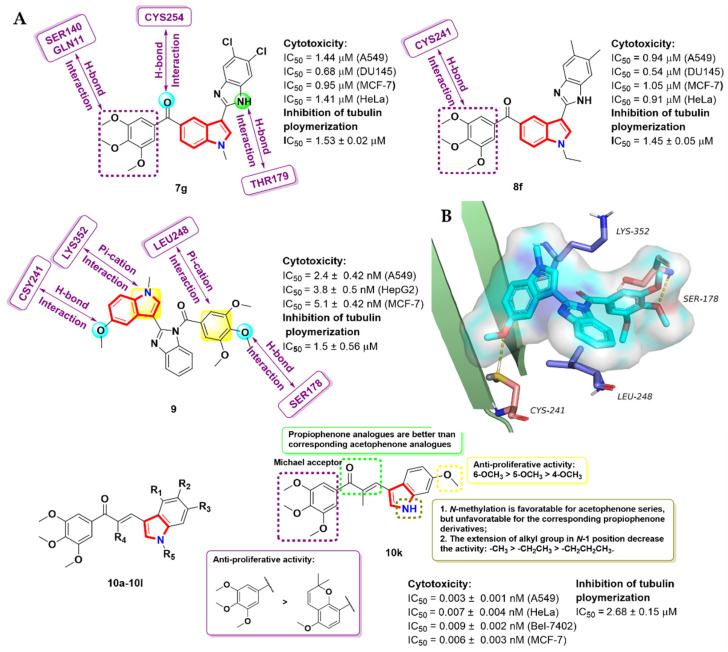
(**A**) Indole-based TMP analogues; (**B**) binding mode for compound **9** with tubulin (PDB code: 1SA0).

**Figure 6 molecules-27-01587-f006:**
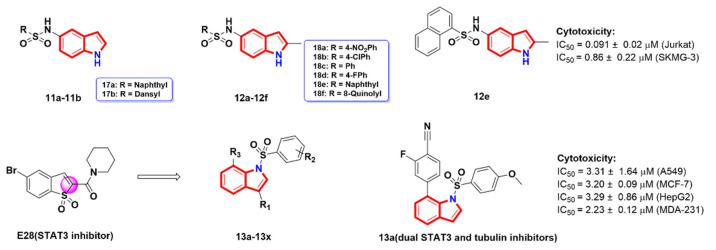
Arythioindole derivatives.

**Figure 7 molecules-27-01587-f007:**
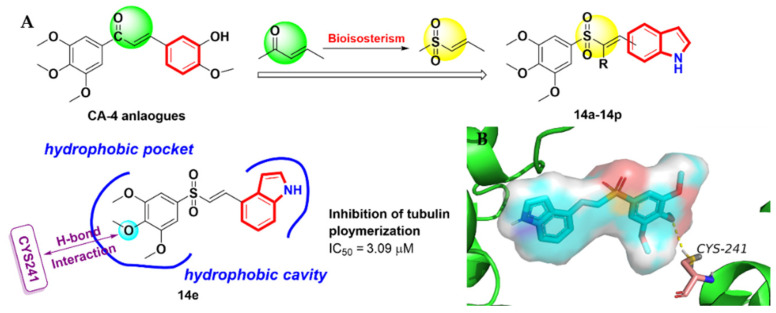
(**A**) Arythioindole derivatives; (**B**) binding mode for compound **14e** with tubulin (PDB code: 1SA0).

**Figure 8 molecules-27-01587-f008:**
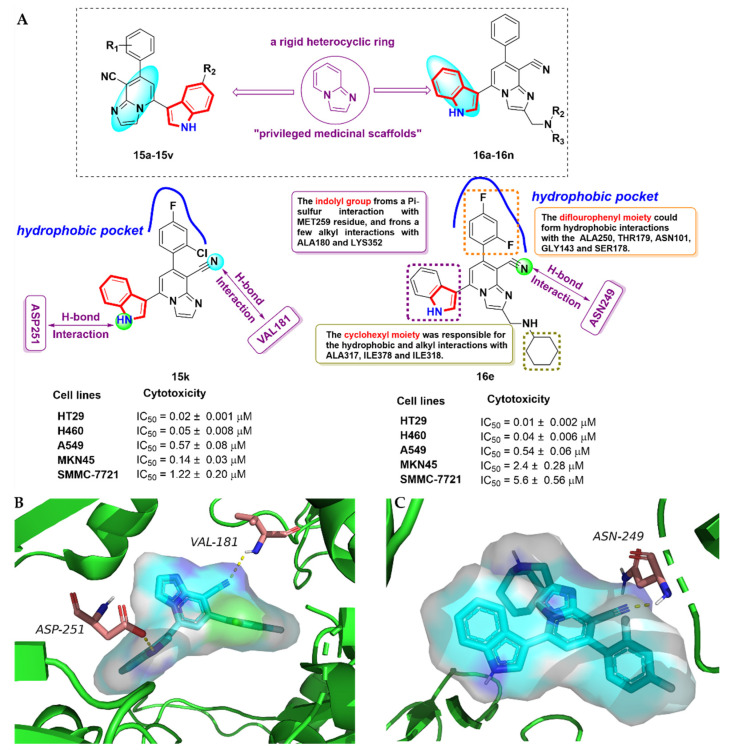
(**A**) Aroyindole derivatives; (**B**) binding mode for compound **15k** with tubulin (PDB code: 4O2B); (**C**) binding mode for compound **16e** with tubulin (PDB code: 4O2B).

**Figure 9 molecules-27-01587-f009:**
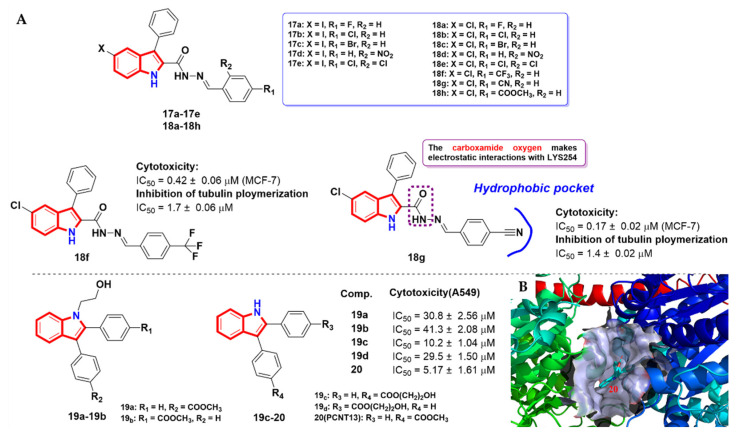
(**A**) Arythioindole derivatives; (**B**) surface model for residues of tubulin and the binding site of compound **20** (PDB code: 1SA0).

**Figure 10 molecules-27-01587-f010:**
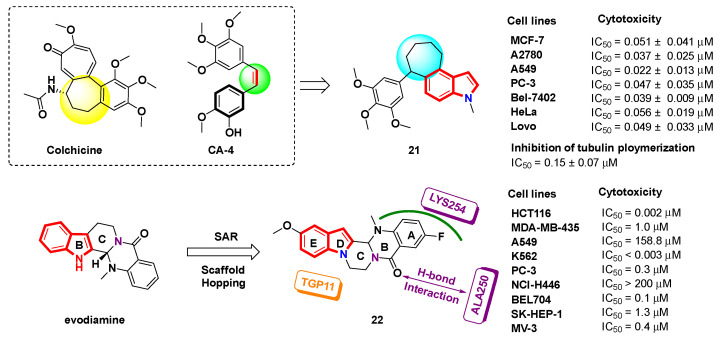
Design strategies for fused indole analogues. Docked structure of compound **22**. (Topoisomerase 1-DNA complex: PDB code: 1T8I, tubulin: PDB code: 1SA0).

**Figure 11 molecules-27-01587-f011:**
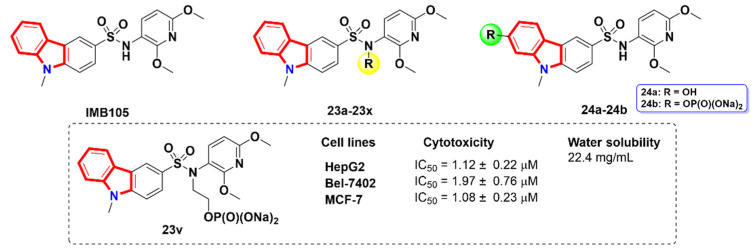
Carbazole derivatives.

**Figure 12 molecules-27-01587-f012:**
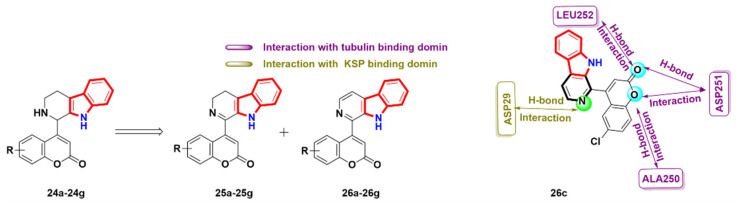
Azacarboline derivatives (tubulin: PDB code: 1SA0, KSP: PDB code: 1Q08).

**Figure 13 molecules-27-01587-f013:**
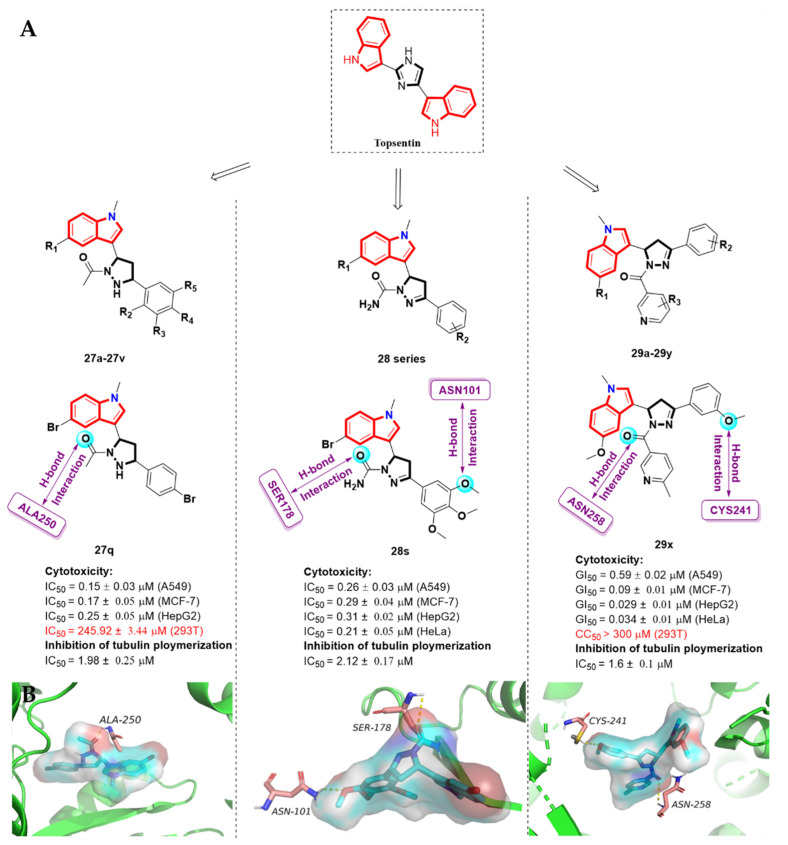
(**A**)Alkaloid nortopsentin analogue derivatives; (**B**) the molecular docking models for compounds **27q**, **28s** and **29x**. (PDB code: 1SA0).

**Figure 14 molecules-27-01587-f014:**
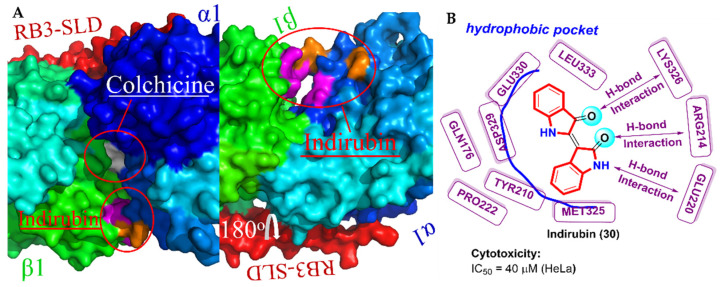
(**A**) Left: In the surface view of tubulin, colchicine (white) bound at the interface of the α_1_-β_1_ (blue and green) tubulin heterodimer. The binding site of indirubin was close to that of colchicine. Right: Three residues on the α_1_ subunit (orange) had hydrogen bond interactions with indirubin (**30**), and the hydrophobic pocket formed by several residues is shown in violet; (**B**) binding mode for indirubin (**30**) with tubulin (PDB code: 1SA0).

**Figure 15 molecules-27-01587-f015:**
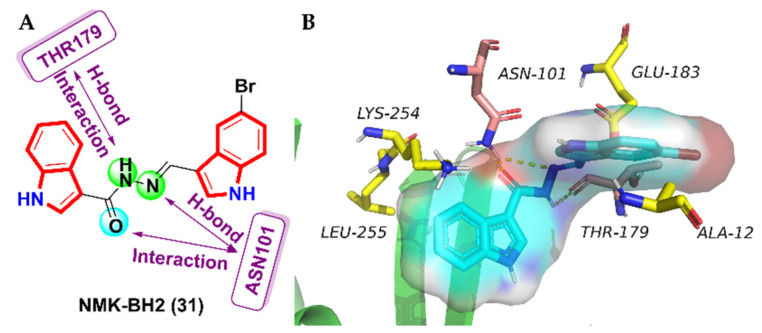
(**A**) Structure of 2-anilinopyridine-arylpropenone analogues; (**B**) binding mode for compound **31** with tubulin (PDB code: 1Z2B).

**Figure 16 molecules-27-01587-f016:**
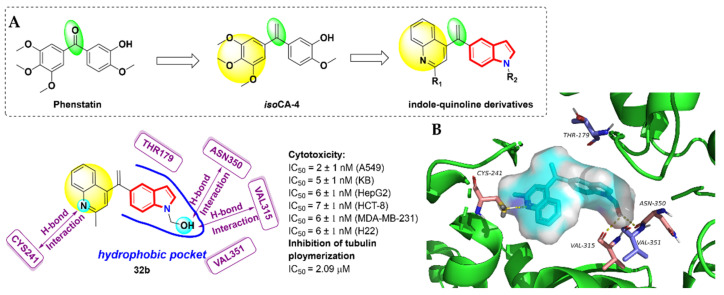
(**A**) Design strategy of novel indole–quinoline derivatives inspired by the structure of known phenstatin and *iso*CA-4. Docked structure of compound **32b**; (**B**) binding mode for compound **32b** with tubulin (PDB code: 5LYJ).

**Figure 17 molecules-27-01587-f017:**
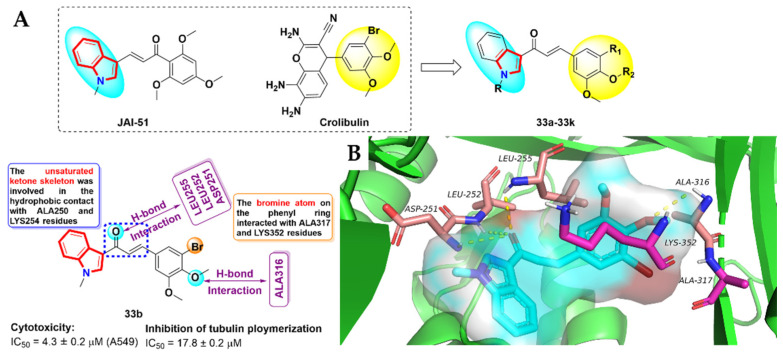
(**A**) Design of novel indole-based compound **33a-33k** from the skeleton of reported tubulin inhibitors JAI-51 and crolibulin. Docked structure of compound **33b**; (**B**) binding mode for compound **33b** with tubulin (PDB code: 1SA0).

**Figure 18 molecules-27-01587-f018:**
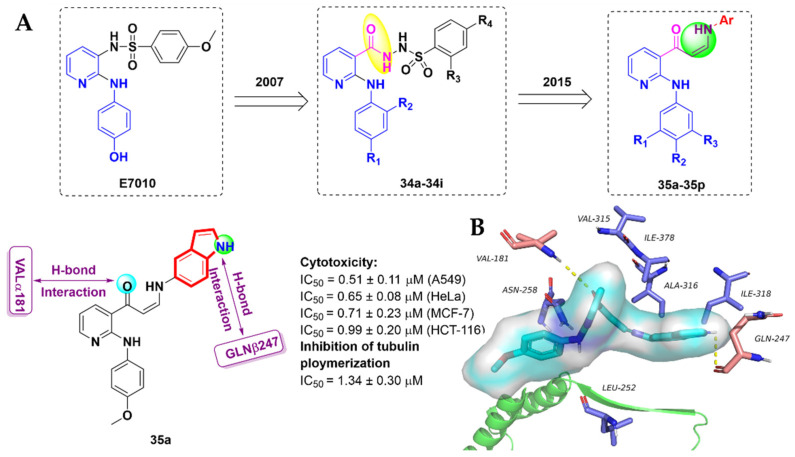
(**A**) Strategy design of novel 2-anilinopyridine-arylpropenone analogues **35a-35p**. Docked structure of compound **35a**; (**B**) binding mode for compound **35a** with tubulin (PDB code: 3UT5).

**Figure 19 molecules-27-01587-f019:**
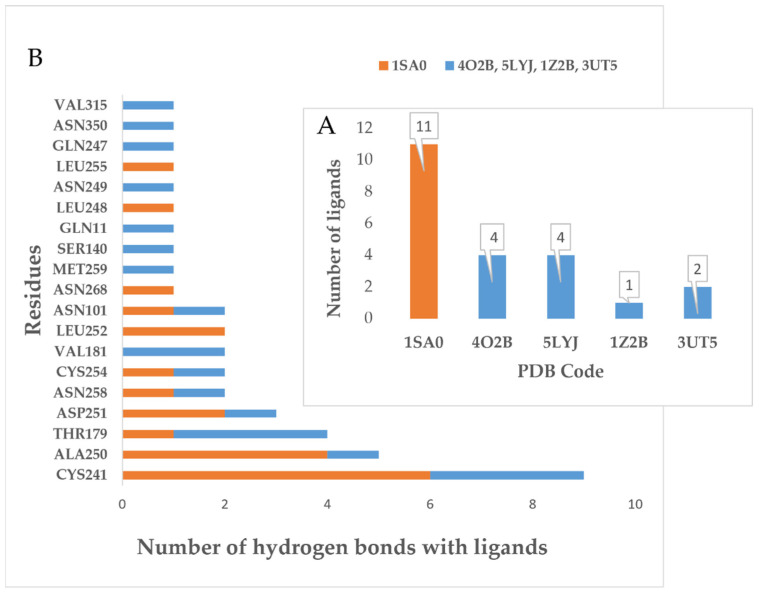
(**A**) All residues that had hydrogen bond interactions with 22 ligands described above (colchicine binding site); (**B**) tubulin proteins used for docking with 22 ligands.

**Figure 20 molecules-27-01587-f020:**
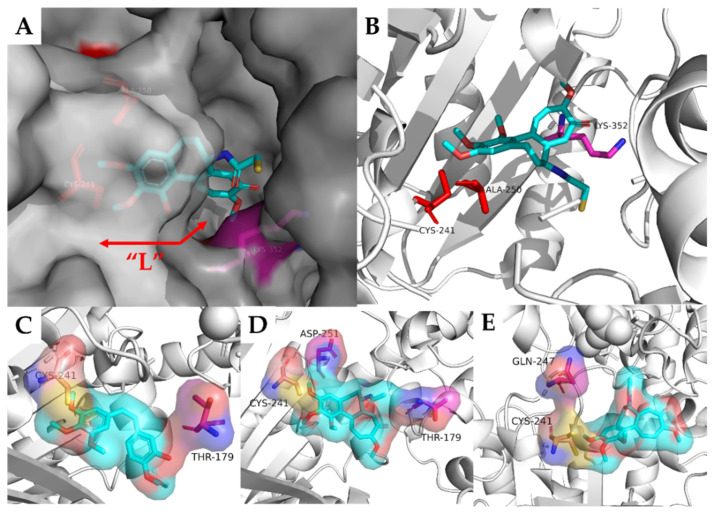
(**A**) In the surface view of tubulin, CYS241 and ALA250 (red) are located in the hydrophobic pocket extended by the TMP group of colchicine. LYS352 (violet) is on the other side of the colchicine “L-shaped” cavity; (**B**) Figure A is shown as cartoon (PDB code: 1SA0); (**C**) THR179 (vol) is adjacent to the “B-ring” side of CA-4 (PDB code: 4LYJ); (**D**) THR179 and ASP251 are close to the seven-member aromatic ring and TMP group of colchicine, respectively (PDB code: 4O2B). (**E**) GLN247 is adjacent to CYS241 and slightly beyond the TMP group (PDB code: 4O2B).

**Table 1 molecules-27-01587-t001:** Some representative indole-based drugs approved by FDA.

Drug Name (Company)	Target	Indications	Year (FDA)
Vincristine (*Eli Lilly*)	Tubulin	Hodgkin lymphoma	1963
Vinorelbine (*Pierre Fabre*)	Tubulin	Hodgkin lymphoma	1994
Alectinib (*Roche*)	ALK	NSCLC	2015
Panobinostat (*Novartis*)	HDACs	Multiple myeloma	2015
Sunitinib (*Pfizer*)	VEGFR	Gastrointestinal stromal tumor	2006
Osimertinib (*AstraZeneca*)	EGFR	NSCLC	2015

**Table 2 molecules-27-01587-t002:** In vitro anti-proliferative activity of indole-vinyl sulfone derivative **14e**.

Comp.	IC_50_ (μM)						
	HepG2	A549	K562	HCT-8	H22	Bel-7402	LO2
**14e**	0.075 ± 0.005	0.305 ± 0.026	0.055 ± 0.005	0.287 ± 0.034	0.060 ± 0.015	0.080 ± 0.023	0.240 ± 0.090

**Table 3 molecules-27-01587-t003:** In vitro anti-proliferative activity of bis(indolyl)-hydrazide-hydrazone derivative **31**.

Comp.	IC_50_ (μM)							
	HeLa	MCF-7	MDA	PA-1	HepG2	PBMC	WI38	HEK
**31**	1.5 ± 0.25	5.25 ± 0.31	8 ± 0.87	24.7 ± 2.61	29.7 ± 3.25	50 ± 4.8	41.5 ± 4.2	46.5 ± 3.5

## Data Availability

Exclude this statemen.

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
