# Peer review of "Indole-Based Tubulin Inhibitors: Binding Modes and SARs Investigations"

_molecules, 2022, doi:10.3390/molecules27051587_

Round 1

Reviewer 1 Report

The authors have done an effort to improve the manuscript, but the new version of this manuscript continues to suffer from the same deficiencies that were reflected in the opinions of the three reviewers on the original.
In the first place, the topic chosen for the realization of this manuscript has been extensively treated in current reviews, as reflected in the search carried out by crossing the terms "tubulin", "inhibition" and "indole", and refining those that were classified as reviews. Ten reviews were found in the period 2016-2022. None of these reviews is mentioned in the manuscript as previous information.
In second place, among these reviews stands out: "Indole derivatives as tubulin polymerization inhibitors for the development of promising anticancer agents. Hong Yu; Zhu Yuan-Yuan; He Qiuqin; Gu Shuang Xi. Bioorganic & medicinal chemistry (2021), 55, 116597". This review contains the same information as that included in the manuscript, but much better organized in structural terms. In this sense, the manuscript does not provide new relevant information that justifies its publication.
In addition, if a search is performed with the aforementioned terms “tubulin” “inhibition” and “indole” and it is refined to the publications of 2020-2021-2022, 33 references are found, most of them in the area of medicinal chemistry. With the exception of that of Romagnoli et al from 2020, none of the others has been reflected in the manuscript, so the information is redundant with that of Yu et al from 2021.
Finally, the novel part of the manuscript could be the docking analysis of the compounds included in it, but this analysis has already been carried out in many cases by the authors themselves in the original publications. Although the study carried out was not without some interest, its conclusion could be summarized in what the authors themselves point out in lines 566-568: “According to Figures 20C-E, we found that the residues (violet) were located in all directions of the cavity, which meant that the orientation of the indole nucleus was also irregular”. As expected, the indole nucleus by itself does not mark the ability to bind to the colchicine site, since this will depend on the rest of the molecule and the global fit of the entire structure in that pocket, so these studies do not lead to any useful conclusion for the design of new ligands. 

Reviewer 2 Report

The revised version of the manuscript has been substantially improved. The authors implemented all of the asked or suggested corrections, so I recommend the manuscript to be accepted for publication.

Reviewer 3 Report

Accept

This manuscript is a resubmission of an earlier submission. The following is a list of the peer review reports and author responses from that submission.

Round 1

Reviewer 1 Report

The manuscript describes a series of antitubulin agents that have indole units in their structure. The manuscript may be of some interest to some researchers interested in this topic, but there are some considerations that discourage its publication in its current version.

1) There is no criteria for the selection of the chosen compounds, having left out of the review compounds that have indole units and that act by inhibiting the polymerization of tubulin (see list of some recent articles that include Tubulin and Indole in the title )

2) The way in which the compounds included in the review have been classified is not explained

3) The article makes a description of the compounds and how some of them bind to tubulin, but no conclusion is drawn from the work carried out or about the advantages that indole brings to tubulin binding.

4) There are some errors, for example in figure 8 the compounds classified as Aroyindoles are included but in the legend Arythioindoles are mentioned

5) English should be reviewed by a native English speaker

Structural insights into the design of indole derivatives as tubulin polymerization inhibitors. FEBS Letters (2020), 594(1), 199-204.

Design, synthesis and biological evaluation of quinoline-​indole derivatives as anti-​tubulin agents targeting the colchicine binding site. European Journal of Medicinal Chemistry (2019), 163, 428-442.

Indirubin, a bis-​indole alkaloid binds to tubulin and exhibits antimitotic activity against HeLa cells in synergism with vinblastine. Biomedicine & Pharmacotherapy (2018), 105, 506-517.

New indole-​based chalconoids as tubulin-​targeting antiproliferative agents. Bioorganic Chemistry (2017), 75, 86-98.

Novel indole-​based compounds that differentiate alkylindole-​sensitive receptors from cannabinoid receptors and microtubules: Characterization of their activity on glioma cell migration. Pharmacological Research (2017), 115, 233-241.

An Orally Bioavailable, Indole-​3-​glyoxylamide Based Series of Tubulin Polymerization Inhibitors Showing Tumor Growth Inhibition in a Mouse Xenograft Model of Head and Neck Cancer. Journal of Medicinal Chemistry (2015), 58(23), 9309-9333.

Reviewer 2 Report

The manuscript (molecules-1560759) entitled " Indole-based tubulin inhibitors: Binding site and cell-based investigation” by Wufu Zhu and co-worker "is a literature review that reports a number of synthetic indole-based tubulin inhibitors and some indole-based alkaloids from natural sources and classifies the results according to their interaction with the binding sites and their biological activity. The computational-based studies and biological studies appear to have been carefully collected. I believe that this manuscript is quite clearly written, but The reviewer recommends MINOR revision of this manuscript for publication, after addressing the following points.

  1. This manuscript lacks a strong conclusion. In fact, the authors have simply selected the library of molecules randomly. A detailed conclusion should be added.
  2. A brief description of the Author’s findings should be added to the manuscript during revision.
  3. The general quality of the figures can be improved.
  4. The reference should be carefully checked. some of them are incorrect (such as reference 7).
  5. Authors have written “Vincristine, a tubulin drug, is an anti-tumor alkaloid ………..” page 2 without citation. Please add the reference.
  6. In the discussion section, the authors should outline some details about the docking protocol e.g. the docking algorithm and the used software, …..
  7. Please, Modify the 3D images in the docking study so that all the amino acids appear clearly.
  8. Still language editing is required to improve the quality of the presentation. In some places, typos errors in the manuscript need to be corrected.
  9. Also, please make sure that all the references are up to date and complete.
  10. The text in the manuscript's main body, should be shortened and further focused for clarity and ease of readability.

Reviewer 3 Report

The review article (molecules-1560759) entitled “Indole-based tubulin inhibitors: Binding site and cell-based in-2 investigation " reports the indole-based tubulin inhibitors, their evaluated inhibitory activity, including active site information. I appreciate the authors' efforts to write a review article covering previous modifications of indole-based tubulin inhibitors. But the area is already well explored, and several review articles have been published recently with similar information (provided the information below). Therefore, this review does not attract the attention of readers.

  1. Indole derivatives as tubulin polymerization inhibitors for the development of promising anticancer agents, Volume 55, 1 February 2022, 116597, https://doi.org/10.1016/j.bmc.2021.116597.

  1. Indole derivatives (2010–2020) as versatile tubulin inhibitors: synthesis and structure–activity relationships, FUTURE MEDICINAL CHEMISTRYVOL. 13, NO. 20, Published Online:1 Sep 2021, https://doi.org/10.4155/fmc-2020-0385.

  1. Indole-based Tubulin Polymerization Inhibitors: An Update on Recent Developments, Mini Rev Med Chem, 2016;16(18):1470-1499. DOI: 10.2174/1389557516666160505115324.